

# The divergence history of European blue mussel species reconstructed from Approximate Bayesian Computation: the effects of sequencing techniques and sampling strategies

Christelle Fraïsse[1,2,3], Camille Roux[4], Pierre-Alexandre Gagnaire[1], Jonathan Romiguier[1], Nicolas Faivre[1,2], John J. Welch[2] and Nicolas Bierne[1,2]

[1] Institut des Sciences de l'Evolution UMR5554, University Montpellier, CNRS, IRD, EPHE, Montpellier, France
[2] Department of Genetics, University of Cambridge, Cambridge, UK
[3] Institute of Science and Technology Austria, Klosterneuburg, Austria
[4] Université de Lille, Unité Evo-Eco-Paléo (EEP), UMR 8198, Villeneuve d'Ascq, France

Corresponding author
Christelle Fraïsse,
christelle.fraisse@ist.ac.at

## ABSTRACT

Genome-scale diversity data are increasingly available in a variety of biological systems, and can be used to reconstruct the past evolutionary history of species divergence. However, extracting the full demographic information from these data is not trivial, and requires inferential methods that account for the diversity of coalescent histories throughout the genome. Here, we evaluate the potential and limitations of one such approach. We reexamine a well-known system of mussel sister species, using the joint site frequency spectrum (jSFS) of synonymous mutations computed either from exome capture or RNA-seq, in an Approximate Bayesian Computation (ABC) framework. We first assess the best sampling strategy (number of: individuals, loci, and bins in the jSFS), and show that model selection is robust to variation in the number of individuals and loci. In contrast, different binning choices when summarizing the jSFS, strongly affect the results: including classes of low and high frequency shared polymorphisms can more effectively reveal recent migration events. We then take advantage of the flexibility of ABC to compare more realistic models of speciation, including variation in migration rates through time (i.e., periodic connectivity) and across genes (i.e., genome-wide heterogeneity in migration rates). We show that these models were consistently selected as the most probable, suggesting that mussels have experienced a complex history of gene flow during divergence and that the species boundary is semi-permeable. Our work provides a comprehensive evaluation of ABC demographic inference in mussels based on the coding jSFS, and supplies guidelines for employing different sequencing techniques and sampling strategies. We emphasize, perhaps surprisingly, that inferences are less limited by the volume of data, than by the way in which they are analyzed.

## INTRODUCTION

The biodiversity we inherited from the Quaternary was shaped by the process of species formation (*Hewitt, 2000*). A long-standing question concerns the timing and rate of gene exchange that occurred while populations diverged, during the incipient stages of speciation. Model-based inferences from genetic data have been used to investigate the history of gene flow (*Beaumont et al., 2010*). Special attention has been paid to the distinction between recent divergence in a strict isolation (SI) model, and older divergence with continuous migration (*Nielsen & Wakeley, 2001*), although of course, more complex scenarios are also possible (*Marino et al., 2013*, *Sousa & Hey, 2013*).

With next-generation sequencing technologies, thousands of SNPs throughout the genome can be used to infer the demographic histories of non-model species pairs (*Sousa & Hey, 2013*). One way of summarizing the information in these data is the unfolded joint site frequency spectrum (jSFS), that is, the number of copies of derived alleles found in each of the two sampled species. A recent and fast maximum-likelihood method based on the jSFS (*Gutenkunst et al., 2009*) has proven useful for distinguishing continuous migration from SI (e.g., in ragworts, *Chapman, Hiscock & Filatov, 2013*, and beach mice, *Domingues et al., 2012*). The method can also evaluate more complex scenarios (e.g., in sea bass, *Tine et al., 2014*; poplars, *Christe et al., 2017*; and whitefish, *Rougeux, Bernatchez & Gagnaire, 2017*), but it struggles to explore the parameter space in these cases. In addition, the method is not well suited for transcriptome data as model comparison by log-likelihood ratio tests assumes independence of SNPs. As a consequence simulations need to be conducted to evaluate competing models, and the computational speed advantage is lost.

As an alternative, Approximate Bayesian Computation (ABC) is a method based on simulations that avoids the need to explicitly compute the likelihood (*Beaumont, Zhang & Balding, 2002*). As such, histories of speciation characterized both by periods of SI and periods of gene exchange can easily be investigated, for example, the scenarios of ancient migration (AM) and secondary contact (SC). These scenarios can be extended by including two cycles of "isolation/gene exchange," following climatic changes in the Pleistocene (Fig. 1). Methods have also been developed to include genome-wide heterogeneity in migration rates (*Sousa et al., 2013*; *Roux et al., 2013*). This is consistent with the "genic view" of speciation (*Wu, 2001*), whereby barriers to gene flow are often semi-permeable, varying in strength across the genome due to linked selection and recombination (*Barton & Bengtsson, 1986*). A major challenge in ABC, as compared to explicit likelihood methods, is the selection of summary statistics, which involves a trade-off between loss of information and reduction of dimensionality. Several methods have been suggested to select the most appropriate statistics for a given dataset and a set of models (*Wegmann, Leuenberger & Excoffier, 2009*; *Nunes & Balding, 2010*; *Aeschbacher, Beaumont & Futschik, 2012*); but most of these summarize the SFS (but see, e.g., *Boitard et al., 2016*) leading to a loss of information. Recently, several ABC studies have used the SFS directly to reconstruct the history of single populations (*Boitard et al., 2016*), or multiple populations (*Xue & Hickerson, 2015*; *Smith et al., 2017*). However, the number of statistics, that is,

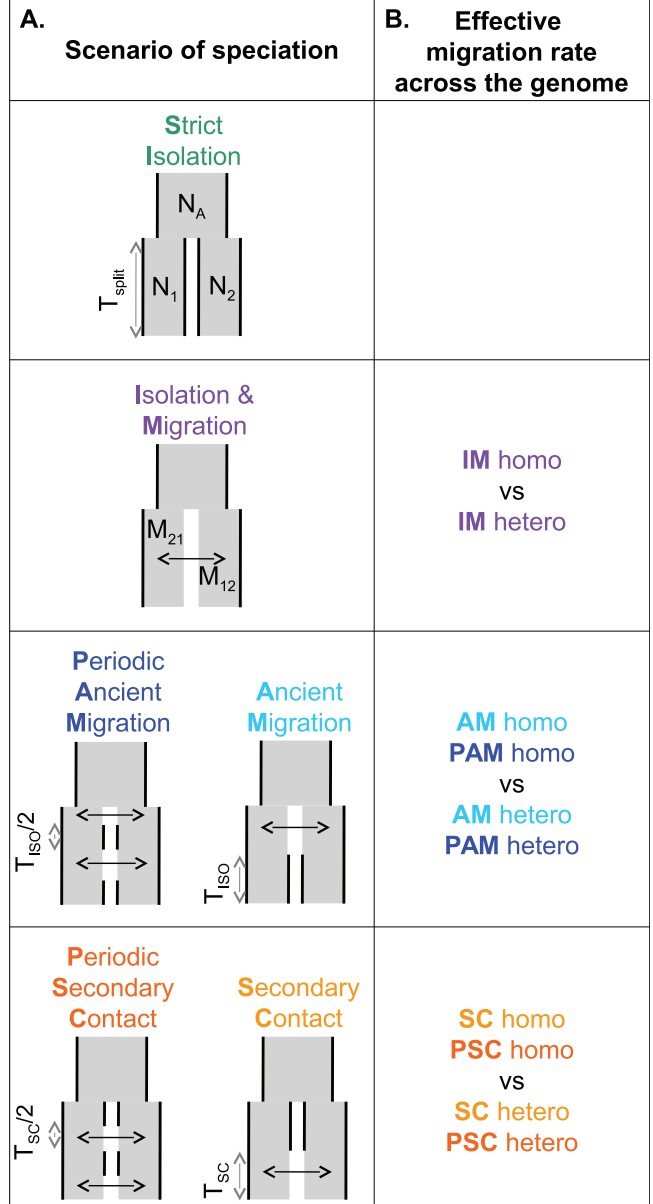

**Figure 1 Models of speciation.** Six classes of scenarios with different temporal patterns of migration are compared (A); and for those including migration, two versions are depicted assuming either homogeneity ("homo") or heterogeneity ("hetero") of effective migration rate across the genome (B). All scenarios assume that an ancestral population of effective size $N_A$ split $T_{split}$ generations ago into two populations of constant sizes $N_1$ and $N_2$. At the two extremes, divergence occurs in allopatry (SI, strict isolation) or under continuous migration (IM, isolation with migration). Through time, migration occurs at a constant rate $M_{12}$ from population 1 to population 2 and $M_{21}$ in the opposite direction. Ancient migration (AM) and periodic ancient migration (PAM) scenarios both assume that populations started diverging in the presence of gene flow. Then they experienced a single period of isolation, $T_{iso}$, in the AM model while intermittent gene flow occurred in the PAM model. In the secondary contact (SC) and periodic secondary contact (PSC) scenarios, populations diverged in the absence of gene flow followed by a single period of secondary contact, $T_{sc}$, in the SC model while intermittent gene flow occurred in the PSC model.

the number of classes in the spectrum, increases quadratically with the number of haploid genomes sampled in the case of a two-dimensional SFS, and even faster if more than two populations are considered. For this reason, recent studies have tested different ways of binning the SFS to circumvent the curse of dimensionality (*Smith et al., 2017*). While accumulating new data and developing new methods, studies that evaluate the impact of the sampling strategy and the inferential method to reconstruct species divergence history will be increasingly valuable (*Li & Jakobsson, 2012*; *Robinson et al., 2014*; *Shafer et al., 2015*; *Cabrera & Palsbøll, 2017*; *Smith et al., 2017*).

*Mytilus edulis* (Linnaeus, 1758) and *M. galloprovincialis* (Lamarck, 1819) are two closely-related species that currently hybridize where their ranges overlap along the Atlantic French coasts (*Bierne et al., 2003a*) and the British Isles (*Skibinski, Beardmore & Cross, 1983*). Their inter-specific barrier to gene flow is semi-permeable, and it has been shown to involve multiple isolating mechanisms, both pre-zygotic (e.g., assortative fertilization and habitat choice; *Bierne et al., 2002*, *Bierne, Bonhomme & David, 2003b*), and post-zygotic (hybrid fitness depression, *Simon, Bierne & Welch, 2017*). Other evidence of ongoing gene flow between *M. edulis* and *M. galloprovincialis* comes from footprints of local introgression of *edulis*-derived alleles into a population of *M. galloprovincialis* enclosed within the Atlantic hybrid zone (*Fraïsse et al., 2014*). Another study (*Fraïsse et al., 2016*) revealed that the Atlantic population of *M. galloprovincialis* was more introgressed than the Mediterranean population on average. At some specific loci, however, the Mediterranean population was found to be fixed for *edulis* alleles while the Atlantic population was not introgressed at all, suggesting that an ancient contact between *M. edulis* and Mediterranean *M. galloprovincialis* occurred during glacial periods. Finally, direct model comparisons have been conducted with the IMa method of *Hey & Nielsen (2007)*, and with an ABC framework, and shown that *M. edulis* and *M. galloprovincialis* have experienced a complex history of divergence punctuated by periods of gene flow in Europe (IMa: *Boon, Faure & Bierne, 2009*, ABC: *Roux et al., 2014*, *2016*).

Here, we analyzed coding sequence datasets from this well-known pair of sister species in Europe, and systematically reconstructed its speciation history by ABC, using different sampling strategies. By different sampling strategies, we mean that we varied (i) the number of individuals sampled (two, four or eight), (ii) the number of SNPs (which were obtained by different sequencing techniques, "exome capture" vs. "rna-seq"), and (iii) the number of bins in the jSFS (binning the spectrum into four, seven or 23 classes). We then evaluated the influence of these choices on model selection, using eleven distinct scenarios of speciation. Our results show that the influence on inferences of the number of individuals and loci sampled is surprisingly limited, while the different ways of binning the jSFS strongly affect the results. This suggests that demographic reconstructions are nowadays more limited by the way data are analyzed than by their volume. Moreover, we find that an history of periodic connectivity, with both ancient and contemporary introgression, and a semi-permeable barrier to gene flow, best fit these data, arguing for the development of flexible inference methods to better describe complex divergence histories (*Simon & Duranton, 2018*).
**Table 1 Sampling design.**

| Technique | *M. edulis* | | | *M. galloprovincialis* | | | *M. trossulus* (outgroup) | | |
|---|---|---|---|---|---|---|---|---|---|
| | Population | Locality | n | Population | Locality | n | Population | Locality | n |
| Exome capture | North Sea | Wadden Sea, Holland | 4 | Brittany | Roscoff, France | 4 | Europe | Tvärminne, Finland | 4 |
| | Bay of Biscay | Lupin/Fouras, France | 4 | Mediterranean Sea | Sète, France | 4 | | | |
| RNA-seq | North Sea | Barfleur, France | 2 | Brittany | Roscoff, France | 2 | USA | Seattle, USA | 1 |
| | Bay of Biscay | La Tremblade, France | 2 | Mediterranean Sea | Sète, France | 2 | | | |

# MATERIALS AND METHODS

## Sampling, sequencing, mapping, and calling

Two datasets were analysed for demographic inferences. They both comprise measures of molecular polymorphism and divergence in coding sequences, obtained for the pair *M. edulis* and *M. galloprovincialis*. Although the adopted sequencing techniques were different among datasets ("exome capture" vs. "rna-seq"), the surveyed populations were similar. The usage of coding sequences to infer divergence histories of closely-related species (e.g., see *Roux et al., 2013*, *2014*, *2016*; *McCoy et al., 2014*; *Li et al., 2017*; *Qi et al., 2017*) is justified for several reasons: (i) synonymous mutations are less affected by direct selection than other categories of mutation, and selection affects chromosomal regions larger than genes themselves, including non-coding regions (e.g., regulatory elements, *Andolfatto, 2005*); (ii) we implicitly modeled the effects of selection against migrant genes by including heterogeneous effective migration rates across the genome.

### Data set 1: "exome capture"

We used the dataset already published in *Fraïsse et al. (2016)* and available on http://www.scbi.uma.es/mytilus/index.php. Briefly, a set of 890 EST contigs was used as a reference for a pre-capture multiplex DNA enrichment in samples of eight individuals from two geographical populations in each species (*M. edulis*: North Sea and Bay of Biscay; *M. galloprovincialis*: Brittany and Mediterranean Sea, Table 1). In addition, we used a sample of four individuals of *M. trossulus* to serve as an outgroup (Table 1). Each DNA library was sequenced twice to increase the per-base coverage (Miseq or GA2X followed by HiSeq2000). After trimming and quality-filtering, reads of each individual were aligned against the same EST reference sequences using the BWA program (bwa-mem, *Li & Durbin, 2009*). Because of the relative divergence between the two species (~2%, Table 2), we adjusted the default parameter of BWA to allow less stringent mapping (minimum seed length $k = 10$ [default: 19], clipping penalty $L = 3$ [5], mismatch penalty $B = 2$ [4], and gap open penalty $O = 3$ [6]). Full methods are described in *Fraïsse et al. (2016)*.

We used a maximum-likelihood method, implemented in the program read2snps (*Tsagkogeorga, Cahais & Galtier, 2012*; *Gayral et al., 2013*), to call genotypes directly from read numbers at each position. The method computes the probability of each possible genotype after estimating the sequencing error rate. To limit bias in the site frequency estimation (*Han, Sinsheimer & Novembre, 2013*), a minimum of 10× coverage, that is,

## Table 2 Summary statistics (mscalc).

| Technique | $n_1$ | $n_2$ | $n_{locus}$ | $n_{SNP}$ | S | S_sd | Sf | Sf_sd | $Sx_1$ | $Sx_1$_sd | $Sx_2$ | $Sx_2$_sd | Ss | Ss_sd |
|---|---|---|---|---|---|---|---|---|---|---|---|---|---|---|
| Exome capture | 2 | 2 | 516 | 3,993 | 7.738 | **6.732** | 0.322 | **1.285** | 3.124 | **3.077** | 3.3 | **3.509** | 0.992 | **1.855** |
|  | 4 | 4 | 557 | 5,092 | 9.142 | **8.076** | 0.097 | **0.583** | 3.555 | **3.434** | 3.896 | **4.006** | 1.594 | **2.482** |
|  | 8 | 8 | 512 | 5,000 | 9.766 | **8.828** | 0.025 | **0.296** | 3.504 | **3.502** | 4.258 | **4.363** | 1.979 | **2.761** |
| rna-seq | 2 | 2 | 2,147 | 17,275 | 8.046 | **6.554** | 0.81 | **3.057** | 2.966 | **3.14** | 2.809 | **2.851** | 1.462 | **2.216** |
|  | 4 | 4 | 1,842 | 17,902 | 9.719 | **7.953** | 0.507 | **2.608** | 3.344 | **3.328** | 3.368 | **3.318** | 2.501 | **3.318** |

| $\pi_1$ | $\pi_1$_sd | $\pi_2$ | $\pi_2$_sd | $\theta_{w1}$ | $\theta_{w1}$_sd | $\theta_{w2}$ | $\theta_{w2}$_sd | $D_1$ | $D_1$_sd | $D_2$ | $D_2$_sd | FST | FST_sd | div | div_sd | netdiv | netdiv_sd |
|---|---|---|---|---|---|---|---|---|---|---|---|---|---|---|---|---|---|
| 0.016 | **0.016** | 0.016 | **0.015** | 0.017 | **0.017** | 0.016 | **0.016** | −0.27 | **0.674** | −0.391 | **0.513** | 0.114 | **0.189** | 0.02 | **0.017** | 0.004 | **0.009** |
| 0.014 | **0.014** | 0.013 | **0.013** | 0.016 | **0.015** | 0.016 | **0.015** | −0.668 | **0.78** | −0.786 | **0.702** | 0.101 | **0.158** | 0.017 | **0.016** | 0.004 | **0.008** |
| 0.012 | **0.012** | 0.012 | **0.012** | 0.017 | **0.015** | 0.019 | **0.016** | −0.942 | **0.78** | −1.143 | **0.67** | 0.088 | **0.141** | 0.015 | **0.015** | 0.003 | **0.008** |
| 0.038 | **0.031** | 0.036 | **0.029** | 0.038 | **0.03** | 0.037 | **0.029** | −0.068 | **0.805** | −0.179 | **0.699** | 0.181 | **0.256** | 0.057 | **0.053** | 0.02 | **0.049** |
| 0.034 | **0.027** | 0.032 | **0.025** | 0.036 | **0.026** | 0.036 | **0.026** | −0.27 | **0.848** | −0.493 | **0.751** | 0.171 | **0.229** | 0.051 | **0.046** | 0.018 | **0.042** |

**Notes:**

*Technique*: rna sequencing ("rna-seq") vs. exome enrichment sequencing ("exome capture"); *n*: number of individuals analyzed in each species; $n_{locus}$: total number of locus $n_{SNP}$: total number of polymorphic sites. The following statistics were calculated for each locus. Their average (in black) and standard deviation (in bold) across all loci are given. *S*: number of polymorphic sites; *Sf*: number of fixed differences; *Sx*: number of exclusive polymorphic sites; *Ss*: number of shared polymorphic sites; π: number of pairwise differences (*Tajima, 1983*); $\theta_w$: Watterson's θ (*Watterson, 1975*); *D*: Tajima's D (*Tajima, 1989a*, *1989b*); FST = $1-\pi_S/\pi_T$: level of species differentiation, where $\pi_S$ is the average pairwise nucleotide diversity within species and $\pi_T$ is the total pairwise nucleotide diversity of the pooled sample across species; div: total inter-specific divergence; netdiv: net molecular divergence measured at synonymous positions.

10 reads per position and per individual, was required to call a genotype. Only genotypes supported at 95% were retained; otherwise missing data was applied. Moreover, paralogous positions were filtered-out using a likelihood ratio test based on explicit modeling of paralogy.

### Data set 2: "rna-seq"

Data set 2 is made of sequenced transcriptomes (RNA-seq) previously generated in a wide meta-analysis study comparing levels of polymorphism across 76 animal species (*Romiguier et al., 2014*). It includes the transcriptomes of four individuals sampled in the same populations as described above and one individual of *M. trossulus* (Table 1). Briefly, for each individual, cDNA libraries were prepared with total RNA extracted from whole body and sequenced on HiSeq2000. Illumina reads (100 bp, paired-end) were mapped with the BWA program on de novo transcriptomes, independently assembled for each species with a combination of the programs Abyss and Cap3, following the strategy B and D in *Cahais et al. (2012)*. Contigs with a per-individual average coverage below ×2.5 were discarded. Genotype calling was performed as described in the first data set using identical filters. Open reading frames were predicted with the Trinity package and sequences carrying no ORF longer than 200 bp were discarded. Full methods are described in *Romiguier et al. (2014)*.

## Data analysis

We restricted our analysis to loci assembled in all individuals, and longer than 300 bp after filtering positions containing missing data, or more than two segregating alleles when the three species were aligned. Only synonymous positions were used. The total number of loci and SNPs retained in each dataset is given in Table 2.
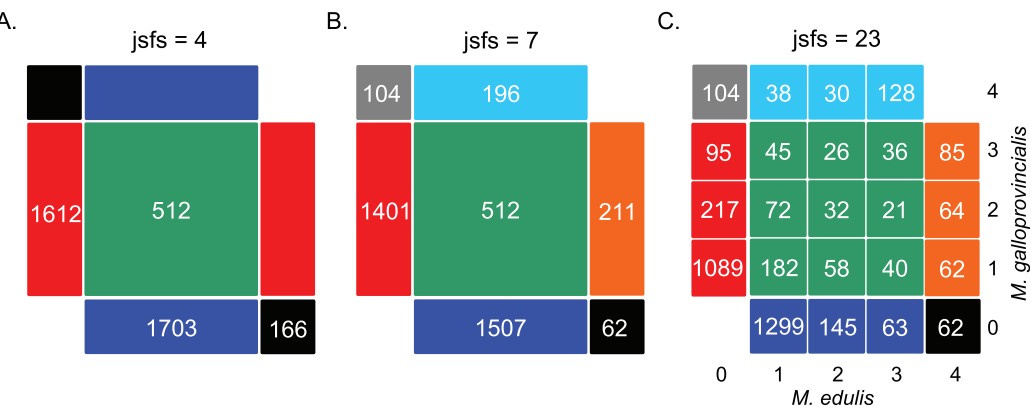

**Figure 2 Decomposition of the unfolded joint site frequency spectrum for $n = 2$ individuals (i.e., four alleles) in each species.** The density of derived alleles in species 1 (*M. edulis*, *x* axis) and species 2 (*M. galloprovincialis*, *y* axis) is indicated by a number within each cell. Only sites showing two distinct alleles in the inter-specific alignment were considered, hence the cells {0; 0} and {4; 4} have been masked. The total number of polymorphic sites is 3,993 SNPs ("exome capture" data). (A) Decomposition of the jSFS into four classes of polymorphism without an outgroup sequence (i.e., the Wakeley–Hey classes): fixed differences (black), private polymorphisms in species 1 (blue) or species 2 (red) and shared polymorphisms (green). (B) Decomposition of the jSFS into seven classes of polymorphism by using the sequenced outgroup. Two alleles are differentially fixed between the two species: the derived allele can be fixed in species 1 (black) or in species 2 (gray). Exclusive polymorphism can be the result of a recent mutation specific to species 1 (blue) or species 2 (red); but it can also be the result of an ancestral mutation only fixed in species 2 (cyan) or in species 1 (orange). Shared polymorphisms are shown in green. (C) Decomposition of jSFS into 23 classes of polymorphism. Singletons and doubletons in each species were included as new classes. Note that in the case of $n = 2$, this is the full spectrum.

## Site frequency spectrum

We first computed the jSFS for each dataset (Fig. 2 and Fig. S1). Each derived allele, oriented by treating the outgroup sequence as a fixed ancestral allele, was assigned to one cell of the jSFS, depending on its frequency in each of the two populations. From the full jSFS, different classes of polymorphism were extracted and used as summary statistics (*Tellier et al., 2011*). Specifically, we used the four Wakeley–Hey classes (jsfs = 4 in Fig. 2): fixed differences, Sf; private polymorphisms for each species $Sx_1$ and $Sx_2$; and shared polymorphisms, Ss (*Wakeley & Hey, 1997*). We also considered a summary in which the Sf and Sx classes were split depending on whether the derived allele was fixed or absent in the other species (jsfs = 7 in Fig. 2; *Ramos-Onsins et al., 2004*). The third decomposition of the jSFS contains 23 classes of polymorphisms because singletons and doubletons in each population were included as new classes (jsfs = 23 in Fig. 2). This corresponds to the full jSFS with $n = 2$ diploid individuals.

## Estimators of polymorphism and divergence

We then computed a set of genetic statistics across loci to make use of the coalescent information contained within each sequence, instead of considering each SNP separately. Following previous studies (*Fagundes et al., 2007*; *Ross-Ibarra et al., 2008*; *Roux et al., 2011*), we used the following statistics: (i) nucleotide diversity, $\pi_1$ and $\pi_2$ (*Tajima, 1983*); (ii) Watterson's $\theta_{w1}$ and $\theta_{w2}$ (*Watterson, 1975*); (iii) total and net inter-specific divergence,

div and netdiv; (iv) between-species differentiation, FST, computed as $1 - \pi_S/\pi_T$, where $\pi_S$ is the average pairwise nucleotide diversity within species and $\pi_T$ is the total pairwise nucleotide diversity of the pooled sample across species. We also included the four Wakeley–Hey's classes as explained above (Sf, $Sx_1$, $Sx_2$, and Ss). Finally, we assessed departure from mutation/drift equilibrium using Tajima's $D_1$ and $D_2$ (*Tajima, 1989a*, *1989b*). The average and standard deviation across the loci of these statistics were calculated with the program MScalc (available from http://www.abcgwh.sitew.ch/, see *Roux et al., 2011*), and their values are given in Table 2 ("mscalc").

## Inferences by Approximate Bayesian Computation

### *Scenarios of speciation*

A total of six distinct scenarios of speciation were considered (Fig. 1). Each scenario modeled an instantaneous division (occurring $T_{split}$ generations ago) of the ancestral population of effective size $N_A$ into two populations of constant sizes $N_1$ and $N_2$. The strict isolation scenario (SI) assumed that divergence occurred without gene exchange between the two populations. The other models differed by their temporal pattern of migration, which occurred at a rate $M_{12}$ from population 1 to population 2, and a rate $M_{21}$ in the opposite direction. Ancient migration and periodic ancient migration (PAM) scenarios both assumed that migration was restricted to the early period of divergence. In the AM scenario, the two populations experienced a single period of SI ($T_{iso}$) while in the PAM scenario, migration was stopped twice with an intermediate period of isolation of $T_{iso}/2$ generations. In the isolation migration (IM), secondary contact, and periodic secondary contact (PSC) scenarios, gene exchange was currently ongoing between the two populations. In the SC and PSC scenarios, the two populations first evolved in SI and then experienced a period of gene exchange ($T_{sc}$). In the SC scenario, there was a single period of recent migration whereas in the PSC scenario a period of ancient migration also occurred after $1 - T_{sc}/2$ generations of SI. The last scenario is the standard IM scenario in which migration occurred continuously over time since the two species started to diverge. For models including migration (IM, AM, PAM, SC, and PSC), we compared two alternative models in which the effective migration rate was either homogeneous ("homo") or heterogeneous ("hetero") among loci (*Roux et al., 2013*, *2014*, *2016*). These models aim to account for the effects of a semi-permeable barrier to gene flow: a reduced effective migration rate is predicted in chromosomal regions linked to incompatible genes; while free introgression is predicted in loosely linked regions.

### *Coalescent simulations*

For each of the five sampling strategies, we performed one million multilocus simulations under the 11 scenarios of speciation using the coalescent simulator Msnsam (*Hudson, 2002*; *Ross-Ibarra et al., 2008*). Simulations fitted to the characteristics of each data set ("exome capture" data with $n = 2, 4, 8$ individuals and "rna-seq" data with $n = 2, 4$ individuals). We assumed free recombination between contigs, and we fixed the intra-contig population recombination rate to be equal to the population mutation rate. Previous studies have shown that methods which take intra-locus recombination into

account remain valid when rates of recombination are low (*Becquet & Przeworski, 2007*). Moreover, our method does not rely on haplotypic data, and so not estimating exact rates of recombination should not affect our results. To account for errors in identifying the ancestral allele in the unfolded jSFS, we explicitly modeled a misorientation rate in our coalescent simulations. We assumed that a proportion $e$ of SNPs, which was a parameter to be inferred, were misoriented and changed $e_i$, their frequency in population $i$, into $1 - e_i$.

Prior distributions for $\theta_A/\theta_{ref}$, $\theta_1/\theta_{ref}$, and $\theta_2/\theta_{ref}$ were uniform on the interval 0–20 with $\theta_{ref} = 4 * N_{ref} * \mu$. The effective size of the reference population, $N_{ref}$, used in coalescent simulations was arbitrarily fixed to 100,000. The mutation rate, $\mu$, was set to $2.763 \times 10^{-8}$ per bp per generation (*Roux et al., 2014*). The $T_{split}/4N_{ref}$ ratio was sampled from the interval 0–25 generations, conditioning the parameters $T_{iso}$ and $T_{sc}$ to be uniformly chosen within the $0 - T_{split}$ interval. For the scenarios including migration, we used the scaled effective migration rates $M = 4Nm$, where $m$ is the fraction of the population made up of effective migrants from the other population at each generation. In the homogeneous model, a single effective migration rate, shared by all loci but differing in each direction of introgression, was sampled from an uniform distribution in the interval 0–40. For the heterogeneous model, we assumed two categories of loci occurring in proportion $p$ and $(1 - p)$. The parameter $p$ was sampled from an uniform distribution in the interval 0–1. The first category of loci are neutral genes introgressing at a migration rate sampled from an uniform distribution in the interval 0–40. The second category comprises loci affected by the barrier to gene flow, so that their effective migration rate was reduced by a gene flow factor (gff), compared to neutral loci (*Barton & Bengtsson, 1986*). The gff was sampled from a $\beta$ distribution with two shape parameters ($\alpha$ chosen in the interval 0.001–10 and $\beta$ chosen in the interval 0.001–5. Prior distributions were computed using a modified version of the program Priorgen (*Ross-Ibarra et al., 2008*) as described in *Roux et al. (2013)*.

### Model choice

To choose the best supported model, we followed the methods previously described in *Roux et al. (2013, 2014)*. Briefly, posterior probabilities for each of the eleven speciation scenario were estimated with a neural network using the R package abc (*Csilléry, François & Blum, 2012*). It implements a non-linear multivariate regression by considering the model itself as an additional parameter to be inferred. The 0.1% replicate simulations nearest to the observed values of the summary statistics were selected. Moreover, to evaluate the relative probability of the heterogeneous model of migration rates across loci, we compared the alternative models ("homo" vs. "hetero") within each scenario including gene exchange. We approximated Bayes factors (BFs) as the posterior probability of the best supported model divided by that of the model with the highest posterior probability from the remaining candidates. The posterior probability of each model calculated among the 11 models of speciation are detailed in Table S1; and those of the best model between heterogeneous and homogeneous migration rates are given in Table S2.

### Model checking

We checked the ability of our ABC framework to correctly recover the true model by a "leave-one-out cross validation" from our simulations. We randomly extracted 100 simulated datasets from the million simulations performed for each model. For each of the 100 of datasets simulated under a given model (i.e., pseudo-observed datasets), we applied the model choice procedure described above to compute the posterior probability of all competitive models. The accuracy rate for model $M$ was calculated as the proportion, among pseudo-observed data inferred to correspond to model $M$, of those actually generated under model $M$. The ambiguity rate was computed as the proportion of pseudo-observed data generated under model $M$ whose best model was not strongly supported, that is, its posterior probability was below an arbitrary threshold ($P_{min}$) set to be one-third above the expected value given the total number of models compared (1/11 for 11 models, 1/6 for six models, or 1/2 for two models). The accuracy and ambiguity rates for the "exome capture" data with $n = 2$ individuals are provided in Table S3.

## RESULTS

### Patterns of polymorphism

We obtained polymorphism data for two mussels species, *M. edulis* and *M. galloprovincialis*, in samples of increasing size ($n = 2$, $n = 4$, and $n = 8$) and for the two datasets ("exome capture" vs. "rna-seq"). The jSFS of each data set, orientated with one outgroup (M. trossulus), is shown in Fig. S1, while Table 2 gives summary statistics of genetic polymorphism.

The data produced by the two sequencing techniques differed in the total number of SNPs ($n_{snp} = 3{,}993$ ($n = 2$); $n_{snp} = 5{,}092$ ($n = 4$); $n_{snp} = 5{,}000$ ($n = 8$) in the "exome capture" data; $n_{snp} = 17{,}275$ ($n = 2$); $n_{snp} = 17{,}902$ ($n = 4$) in the "rna-seq" data). The substantially lower number of SNPs in the "exome capture" data reflects the lower number of loci retained for the analysis due to a reduced and more heterogeneous sequencing depth compared to the "rna-seq" data (while applying identical coverage filters to call SNPs). However, the jSFS calculated from the two techniques had a similar proportion of sites in the different classes (Pearson's correlation between jSFS: $R^2 = 0.97$, $p < 0.0001$ for $n = 2$ individuals; and $R^2 = 0.99$, $p < 0.0001$ for $n = 4$ individuals). Globally, the mean level of genetic differentiation was quite low (FST ~10% in the "exome capture" data; FST ~18% in the "rna-seq" data, see Table 2), but highly variable across loci (standard deviation was ~16% and ~24%, respectively). A low proportion of sites (<5% in the "exome capture" data and <10% in the "rna-seq" data) were fixed differences, while two to three times more SNPs were polymorphic and shared between the two species (see Fig. S1). The level of intra-specific nucleotide diversity was elevated ($\pi_{edu} = \pi_{gal} = 0.016$ in the "exome capture" data; $\pi_{edu} = 0.038$ and $\pi_{gal} = 0.036$ in the "rna-seq" data for $n = 2$, Table 2) and not significantly different between the two species (non-significant Wilcoxon signed-rank test). Polymorphic sites were mainly private to each species (~80% of the sites), and mainly corresponded to low frequency classes. Moreover, the jSFS was

remarkably symmetric suggesting limited differences in population size and/or migration rates between the two species. These patterns were consistent across sample sizes, but there were some differences comparing the two techniques. Specifically, the "exome capture" data set showed significantly lower level of divergence (Wilcoxon signed-rank test, $p < 0.0001$ between $netdiv_{capture} = 0.004$ and $netdiv_{rna-seq} = 0.02$ for $n = 2$ individuals, Table 2) and average number of fixed differences between species per locus (Wilcoxon signed-rank test, $p < 0.0001$ between $Sf_{capture} = 0.322$ and $Sf_{rna-seq} = 0.810$ for $n = 2$ individuals, Table 2). These discrepancies were most likely due to the use of a single reference in the "exome capture" data resulting in the problematic mapping of highly divergent alleles from the two species.

## Effects of the number of individuals and SNPs on model selection

We carried out model selection between the various scenarios of speciation shown in Fig. 1, and asked whether the number of individuals and number of SNPs had an effect. Results are shown in Table 3A, which reports the posterior probability of the best supported scenario for each sampling strategy; and in Table 3B which compares the posterior probability of homogeneous vs. heterogeneous migration for the best supported scenario (see also Tables S1 and S2 for full details).

Firstly, we compared the "exome capture" and the "rna-seq" data which differ in the number of SNPs sampled (3,993 and 17,275 SNPs, respectively); and we found that the best supported scenario was the same for both data sets (Table 3A). For example, when considering 23 classes in the jSFS (jsfs = 23), the best supported scenario always involved recent and genome-wide heterogeneous migration between the two species. The heterogeneous PSC was the most supported scenario; and the next best model described a very similar history of SC with a single period of gene exchange: $BF = P_{PSC.hetero}/P_{SC.hetero} = 1.21$ with the "exome capture" data, $BF = P_{PSC.hetero}/P_{SC.hetero} = 1.07$ with the "rna-seq" data, in the case of $n = 2$ individuals. With $n = 4$ individuals, for which the next best model also included gene flow, those numbers were $P_{PSC.hetero}/P_{IM.hetero} = 1.09$ and $P_{PSC.hetero}/P_{IM.hetero} = 1.37$. The same patterns were found when using different subsets of the jSFS (jsfs = 4 and jsfs = 8, Table 3A), except that the best supported scenario was then the PAM. Regarding genome-wide heterogeneity of migration rates (Table 3B), the "rna-seq" data gave more support to the heterogeneous model (e.g., $BF = P_{AM.hetero}/P_{AM.homo} = 1.25$ with $n = 2$ and jsfs = 4) compared to the "exome capture" data ($BF = P_{AM.homo}/P_{AM.hetero} = 1.26$). This is consistent with the higher heterogeneity of the jSFS in the "rna-seq" data, involving a higher proportion of fixed differences and shared polymorphic sites (Table 2 and Fig. S1).

Secondly, we evaluated the effect of the number of individuals sampled. As with the number of SNPs, it is clear that sample size had little effect. The best supported scenario remained consistent across the different sampling size ($n = 2, 4,$ or $8$ individuals; Table 3A). For example, when considering 23 classes in the jSFS (jsfs = 23), the heterogeneous PSC scenario was the best supported model whatever the sampling size in both datasets.

**Table 3 Posterior probabilities of the speciation models.**

**(A) 11 models**

| Technique | Statistics | n = 2 Scenario | PP | BF$_{1/2}$ | BF$_{1/3}$ | n = 4 Scenario | PP | BF$_{1/2}$ | BF$_{1/3}$ | n = 8 Scenario | PP | BF$_{1/2}$ | BF$_{1/3}$ |
|---|---|---|---|---|---|---|---|---|---|---|---|---|---|
| Exome capture | jsfs = 4 | PAM Homo | 0.360 | 1.45 | 1.87 | PAM Homo | 0.340 | 1.33 | 1.70 | PAM Homo | 0.334 | 1.14 | 2.15 |
| | jsfs = 7 | PAM Homo | 0.309 | 1.26 | 1.34 | PAM Homo | 0.250 | 1.17 | 1.21 | PAM Homo | 0.337 | 1.33 | 2.15 |
| | jsfs = 23 | PSC Hetero | 0.385 | 1.21 | 1.53 | PSC Hetero | 0.363 | 1.09 | 1.59 | PSC Hetero | 0.609 | 2.51 | 4.61 |
| rna-seq | jsfs = 4 | PAM Hetero | 0.227 | 1.14 | 1.60 | PAM Hetero | 0.180 | 1.08 | 1.14 | – | – | – | – |
| | jsfs = 7 | PAM Hetero | 0.323 | 1.62 | 2.27 | PAM Hetero | 0.339 | 1.43 | 2.15 | – | – | – | – |
| | jsfs = 23 | PSC Hetero | 0.414 | 1.07 | 2.60 | PSC Hetero | 0.346 | 1.37 | 1.47 | – | – | – | – |

**(B) Homo vs. hetero for the best model**

| n = 2 Scenario | PP | BF$_{1/2}$ | n = 4 Scenario | PP | BF$_{1/2}$ | n = 8 Scenario | PP | BF$_{1/2}$ |
|---|---|---|---|---|---|---|---|---|
| Homo | 0.557 | 1.26 | Homo | 0.538 | 1.16 | Homo | 0.509 | 1.04 |
| Homo | 0.695 | 2.27 | Homo | 0.519 | 1.08 | Homo | 0.551 | 1.23 |
| Hetero | 0.990 | 99 | Hetero | 0.982 | 54 | Hetero | 1.000 | NA |
| Hetero | 0.556 | 1.25 | Hetero | 0.575 | 1.35 | – | – | – |
| Hetero | 0.733 | 2.75 | Homo | 0.520 | 1.08 | – | – | – |
| Hetero | 0.818 | 4.49 | Hetero | 0.997 | 323.33 | – | – | – |

**Notes:**

$n$: number of individuals analyzed in each species; *technique*: rna sequencing ("rna-seq") vs. exome enrichment sequencing ("exome capture"); statistics: jsfs = 4 (four classes), jsfs = 7 (seven classes), jsfs = 23 (23 classes); PP: posterior probability; BF$_{1/2}$: Bayes factor defined as PP$_{best model}$/PP$_{Second best model}$; BF$_{1/3}$: Bayes factor defined as PP$_{best model}$/PP$_{Third best model}$. 11 models: SI, IM hetero, IM homo, AM hetero, PAM homo, PAM hetero, SC homo, SC hetero, PSC hetero, PSC homo. Colors match Fig. 1.

## Effects on model selection of the number of classes in the jSFS

We next investigated the effects of binning the jSFS: (i) on model choice in the mussel datasets (Fig. 2; Table 3) and (ii) on the ability of the method to discriminate between different speciation scenarios based on simulated datasets (Table S3). Given the limited effects of the sampling strategy, the ABC performance results are presented for a simulated dataset resembling the "exome capture" data with $n = 2$ individuals only. This allowed us consider the full jSFS when using 23 classes of polymorphism.

### Heterogeneity of migration rates

The scenarios with recent migration (SC, PSC, and IM) all strongly supported heterogeneity of migration rates; and this support tended to increase with the number of polymorphic classes considered (Table S2). For example in the "exome capture" data with $n = 2$ individuals, the relative probability of the heterogeneous model in the PSC scenario was $P_{PSC.hetero} = 0.53$ with jsfs = 4, $P_{PSC.hetero} = 0.77$ with jsfs = 7 and $P_{PSC.hetero} = 0.99$ with jsfs = 23. In contrast, the model of homogeneous migration rates was the most supported, though marginally, in the scenarios of PAM and AM: for example, $P_{PAM.homo} = 0.56$ with jsfs = 4, $P_{PAM.homo} = 0.69$ with jsfs = 7 and $P_{PAM.homo} = 0.63$ with jsfs = 23. Concordant patterns were obtained using the other jSFSs, which differed in the number of individuals and SNPs sampled (Table S2).

Using simulated datasets, we then assessed the performance of the method in identifying the correct model when homogeneous and heterogeneous models were compared (Table S3A). The correct model was always recovered (i.e., an accuracy rate of 1) for the different binnings in all speciation scenarios including gene flow; however, the ambiguity rate did strongly decrease when more information from the jSFS was included. With only four classes (jsfs = 4), none of the replicates showed a posterior probability higher than 0.83 (the threshold set for the two-model comparison), which corresponds to an ambiguity rate of 1. Similarly, all ambiguity rates were above 0.97 with seven classes (jsfs = 7). In contrast, when considering the full jSFS (jsfs = 23), we could correctly recover with a strong support the simulated speciation models (e.g., 63% of the PSC.hetero replicates and 43% of the PAM.hetero replicates were above the threshold, Table S3A). These results suggest that the additional classes of the 23-binned jSFS are necessary to detect heterogeneity of migration rates across the genome.

## Scenarios of speciation

It is clear from Table 3A (see details in Table S1) that binning the jSFS to four or seven classes leads to a loss of information. Specifically, when considering the full jSFS (jsfs = 23), only the scenarios involving recent migration were supported ($P_{PSC} + P_{SC} + P_{IM} = 0.96$, Table S1); while the contrary was true when fewer classes of polymorphism were used ($P_{PAM} + P_{AM} = 0.96$ with jsfs = 4 and 0.93 with jsfs = 7, Table S1). Remarkably the SI scenario was never supported ($P_{SI} < 0.06$, Table S1) suggesting that gene flow must have occurred between the two mussels species during divergence. Moreover, the fact that the most supported scenario, using full information (jsfs = 23), was the heterogeneous PSC ($P_{PSC.hetero} = 0.39$) suggests a complex history of speciation, including periods of

isolation alternating with ancient and recent migrations. The discrepancies that appear when not distinguishing low and high frequency shared variants (in the case of jsfs = 4 and jsfs = 7), confirm their importance for the identification of recent migration events (*Alcala et al., 2016*). In general, the best supported model fits well the data for each binning strategy (Fig. S2). All observed statistics were in the 95% simulated posterior distribution, except for the class "ssfB_2" (derived polymorphism fixed in species B, but polymorphic (doubletons) in species A) which was slightly overestimated by the model PSC.hetero (jsfs = 23, Fig. S2A); and the class "sfB" (derived polymorphism fixed in species B, and absent in species A) which was slightly underestimated by the model PAM.hetero (jsfs = 7, Fig. S2B). No statistics were found significantly out of the simulated distribution under model PAM.hetero with jsfs = 4 (Fig. S2C). On the contrary, the alternative models failed to fit more parts of the jSFS. For example, with jsfs = 23 (Fig. S2A), in which PSC.hetero is the best model ($P_{PSC} = 0.385$), the alternative model PAM.hetero ($P_{PAM} = 0.007$) overestimates the number of derived sites fixed within each species in all "sf" classes and model SI ($P_{SI} = 0.009$) underestimates the number of shared polymorphisms in most "ss" classes.

We further evaluate, by simulation, the effect of binning the jSFS on the capacity of the method to infer the correct speciation model (Table S3B). As the heterogeneous models consistently outperformed the homogeneous models in scenarios with ongoing migration, and they were not significantly less likely in the models of ancient migration (Table S2), the ABC performance was evaluated among the five models of migration including heterogeneous migration only (IM.hetero, SC.hetero, PSC.hetero, AM.hetero, and PAM.hetero), plus the SI model. Globally, the probability of rejecting the correct model decreased when increasing the number of polymorphic classes. By using four or seven classes vs. 23 classes, the correct model was recovered at an estimated rate equal to or lower than 0.70 vs. 1 for IM.hetero, 0.58 vs. 0.50 for SC.hetero, 0.89 vs. 1 for PSC.hetero, 0.34 vs. 0.42 for AM.hetero, 0.70 vs. 0.79 for PAM.hetero, and 0.73 vs. 0.84 for SI (Table S3B). Moreover, we could not discriminate between the different scenarios of recent migration (PSC.hetero, SC.hetero, and IM.hetero) when using only four or seven classes (Fig. S3A); and the same was true with the two scenarios of PAM.hetero and AM.hetero (Fig. S3B). In contrast, the full jSFS (jsfs = 23) contains enough information to more accurately identify the PSC as the true model among the other scenarios of recent migration (Fig. S3A). Nevertheless, distinguishing between the AM and PAM scenarios remained difficult even with jsfs = 23 (Fig. S3B). Finally, the ambiguity rates also strongly decreased when binning the jSFS in more classes: all ambiguity rates were equal to or greater than 0.86 in the recent migration scenarios, 0.41 in the ancient migration scenarios and 0.21 in the SI scenario with jsfs = 4 or jsfs = 7; whereas these numbers were 0.25, 0.20, and 0.15 with jsfs = 23 (Table S3B).

## DISCUSSION

NGS data give us the opportunity to capture the diversity of coalescent histories across loci, and so to reveal the complexity of the speciation process (*Sousa & Hey, 2013*). Recently, important efforts have been made to develop statistical methods of inference

making use of population genomics data. Computing the jSFS is an efficient way of summarizing the demographic information contained in NGS data because anonymous SNPs can be used (e.g., produced by RAD-sequencing) and it does not rely on phased data. However, the jSFS obtained from low-coverage sequencing data (typically <10× per position and per individual) can be biased toward a deficit of rare variants; and this is of particular concern when investigating the demographic history of populations (*Nielsen et al., 2012*; *Han, Sinsheimer & Novembre, 2013*). A first category of maximum-likelihood methods uses forward diffusion theory to compute numerical solutions to the jSFS under complex models (see *Gutenkunst et al., 2009*; *Lukic & Hey, 2012*). A second category of methods estimates the expected jSFS under any demographic model, as simulations are used to approximate the composite-likelihood of the data (see *Naduvilezhath, Rose & Metzler, 2011*; *Excoffier et al., 2013*). Here, we used a likelihood-free method (ABC) on the jSFS of coding sequences, and we evaluated the influence of different sampling strategies on the inference of speciation models in mussels.

Our ABC-based model comparison shows little qualitative effect of individual and SNP sampling on the outcomes. First, inferences based on $n = 2$, $n = 4$, or $n = 8$ individuals supported the same model of PSC (Table 3), consistent with previous results that relatively few individuals are sufficient to make robust inferences on the way divergence occurs between lineages (*Robinson et al., 2014*). This is because most coalescence events occur recently in the population history; so increasing the number of individuals is only helpful to characterize recent demographic events rather than the past divergence history. Second, we showed that inferences from the two sequencing techniques ("exome capture" vs. "rna-seq") were qualitatively the same, despite the very different number of SNPs that were sampled, again supporting the PSC scenario (Table 3). This implies that neither the sequencing technique, nor the number of informative sites have a substantial effect on the inference. In fact, the jSFS calculated from the different data sets were very similar (Fig. S1); we only found a deficit of divergent SNPs in the "exome capture" data that may be due to the difficulty of mapping highly divergent alleles onto a single reference (on the contrary, the "rna-seq" reads from the different species were independently assembled and mapped). From a theoretical perspective, adding more loci (or longer loci) provide information about deep coalescent events which are important for shedding light on the divergence history of closely-related species (*Wang & Hey, 2010*). In fact, previous simulation studies showed an influence of locus length and locus number on the ABC performance in model choice, and highlighted a threshold effect above which adding more loci did not significantly improve inferences (*Li & Jakobsson, 2012*; *Robinson et al., 2014*; *Shafer et al., 2015*). Thereby, we argue that the number of SNPs sampled in our study was sufficient to accurately represent the diversity of coalescent histories in the *Mytilus* genome, and consistently support the same speciation model.

In most studies, functions of the jSFS are used as summaries of the data. For example, the likelihood method of *Nielsen & Wakeley (2001)* uses four classes of polymorphisms to estimate migration rates and divergence times in the IM scenario. In the ABC approach, choosing a suitable set of summary statistics is difficult because it implies a trade-off between loss of information and reduction of dimensionality. Accordingly, practical

methods to identify approximately sufficient statistics have been developed (*Wegmann, Leuenberger & Excoffier, 2009*; *Nunes & Balding, 2010*; *Aeschbacher, Beaumont & Futschik, 2012*). Here, we compared ABC inferences based on 23-binned jSFS (jsfs = 23) with inferences using a subset of polymorphic classes (jsfs = 4 and jsfs = 7). Model checkings through simulations pointed out the loss of information when only four or seven classes in the jSFS were considered; particularly for the inference of recent migration events. By decomposing the jSFS into 23 classes, we could reveal the excess of shared polymorphisms that are at high frequency in one species and low frequency in the other, a pattern produced by recent migrants (Table 3). This is in agreement with the simulation study of *Tellier et al. (2011)* that showed a significant improvement in the estimation of the timing of gene flow when these additional polymorphic classes were considered; and the study of *Alcala et al. (2016)* that showed an excess of high frequency derived alleles is the characteristic footprint of secondary contacts. Across all models, we showed that the probability to correctly infer the true model (accuracy rate) increases with the number of classes considered, and the ambiguity rate correspondingly decreases (Table S3). *Smith et al. (2017)* similarly found that the statistical power of their ABC model selection increases with the number of classes until reaching a plateau of error rates (but above which computation efforts continues to increase).

Inferences based on the 23-bin jSFS consistently support a model of PSC with genomic heterogeneity in gene flow (PSC.hetero) in both the "exome capture" and "rna-seq" mussel datasets, and simulations showed that this model has a high accuracy rate, and one of the lowest ambiguity rates (Table S3). Although its relative posterior probability was moderately higher than that of the other models with migration (SC and IM, Table S1), we showed that the method has some power to distinguish PSC from these models (Fig. S3A). These results are in agreement with previous ABC-based studies revealing clearly a secondary contact history between the two mussel species (*Fraïsse et al., 2014*; *Roux et al., 2014*), although they relied on eight nuclear loci and on a different set of summary statistics (those that we called "mscalc" here). The periodic connectivity models (PSC and PAM) were not included in these previous studies because of the lack of power to test for intermittent gene exchange since secondary contact. In the present study, we directly compared the use of the jSFS vs. "mscalc" statistics by performing additional ABC inferences on the "mscalc" statistics presented in Table 2. Results were similar to those obtained with jsfs = 23 (Table S1), that is, the best supported models included one or two SCs, except for two datasets ("exome capture" data with $n = 2$ and $n = 8$ individuals) for which SI was chosen with a posterior probability of 37% and 34%, respectively. However, the goodness-of-fit of the SI model was quite poor for the standard deviation of the number of fixed sites, "sf_std," and very poor for the standard deviation of the FST, "fst_std" ("exome capture" data with $n = 2$ individuals, Fig. S2D). Both statistics were underestimated by the model suggesting that a history without gene flow cannot produce the observed variation of genetic divergence along the genome. Across models, inferences based on the "mscalc" statistics showed similar accuracy rate to those based on jsfs = 23; however, the ambiguity rates for models with current migration were somewhat higher (PSC = 0.52 vs. 0.30, SC = 0.61 vs. 0.31, IM = 0.40 vs. 0.25, respectively). These

supplementary analyses suggest that extracting summary statistics from the jSFS can lead to a substantial loss of information.

## CONCLUSION

In this work, we have shown that two high-throughput sequencing datasets ("exome capture" and "rna-seq"), imply the same history of divergence in mussels, regardless of the number of individuals or SNPs sampled, but conditional on the inclusion of informative classes in the jSFS. Thus, genome-wide data coupled with flexible inference methods allow us to test for more complex scenarios of divergence, by providing a comprehensive picture of the gene histories across the genome. Here, we incorporate in our ABC framework heterogeneity in migration rates among loci, to account for the semi-permeability of the barrier to gene flow between recently diverged species (*Barton & Bengtsson, 1986*). This variation in the rate of effective migration results in variation of genetic divergence along the genome. As shown elsewhere (*Roux et al., 2013*, *2014*, *2016*), failing to account for this heterogeneity can mislead inferences. In a similar way, the effect of background selection (i.e., purifying selection at linked loci) and genetic hitchhiking (i.e., positive selection at linked loci) in regions of low recombination can now be incorporated in demographic inferences by including heterogeneity in effective sizes among loci (*Sousa & Hey, 2013*, *Roux et al., 2016*, *Aeschbacher et al., 2017*). Combining modelling approaches that account for both sources of genomic heterogeneity (e.g., *Roux et al., 2016*) may provide further insight into the complex interplay between linked selection and resistance to introgression during speciation with gene flow (e.g., *Duranton et al., 2018*).

## ACKNOWLEDGEMENTS

The computations were performed at the "Montpellier Bioinformatics Biodiversity" computing cluster platform (CeMEB LabEx "Mediterranean Center for Environment and Biodiversity").

### Funding

This work was funded by the Agence Nationale de la Recherche (HYSEA project, ANR-12-BSV7-0011), a Languedoc-Roussillon Region "Chercheur(se)s d'avenir" grant (Connect7 project) and the ANR grant CoGeDiv ANR-17-CE02-0006-01. The funders had no role in study design, data collection and analysis, decision to publish, or preparation of the manuscript.

### Grant Disclosures

The following grant information was disclosed by the authors:
Agence Nationale de la Recherche (HYSEA project, ANR-12-BSV7-0011).
Languedoc-Roussillon Region "Chercheur(se)s d'avenir" grant (Connect7 project).
ANR grant CoGeDiv ANR-17-CE02-0006-01.

## Competing Interests

The authors declare that they have no competing interests.

## Author Contributions

- Christelle Fraïsse conceived and designed the experiments, performed the experiments, analyzed the data, prepared figures and/or tables, authored or reviewed drafts of the paper, approved the final draft.
- Camille Roux conceived and designed the experiments, contributed reagents/materials/ analysis tools, authored or reviewed drafts of the paper, approved the final draft.
- Pierre-Alexandre Gagnaire conceived and designed the experiments, contributed reagents/materials/analysis tools, approved the final draft.
- Jonathan Romiguier conceived and designed the experiments, contributed reagents/ materials/analysis tools, approved the final draft.
- Nicolas Faivre conceived and designed the experiments, contributed reagents/materials/ analysis tools, approved the final draft.
- John J. Welch conceived and designed the experiments, authored or reviewed drafts of the paper, approved the final draft.
- Nicolas Bierne conceived and designed the experiments, authored or reviewed drafts of the paper, approved the final draft.

## Data Availability

The scripts used for ABC analysis are provided in the Supplemental Files.

Data set 1: "exome capture": http://www.scbi.uma.es/mytilus/index.php (*Fraïsse et al., 2016*).

Data set 2: "rna-seq": http://kimura.univ-montp2.fr/PopPhyl (*Romiguier et al., 2014*).

## Supplemental Information

Supplemental information for this article can be found online at http://dx.doi.org/10.7717/ peerj.5198#supplemental-information.

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
