# Peer review of "The divergence history of European blue mussel species reconstructed from Approximate Bayesian Computation: the effects of sequencing techniques and sampling strategies"

_PeerJ, doi:10.7717/peerj.5198_

## Round 0.1 · original submission · Major Revisions

Thank You for submitting your work for consideration with PeerJ.

I agree with the referees that your work may be acceptable for publication. The main issue in the current version is "overstating" the generality of your conclusion, manly because; (a) you do not know the "truth" although your interpretation seems to rely heavily upon believing that you have an idea of what went on. The only way to know for sure is with simulated data.

Also your findings mainly apply to your species and care should be taken to not over-generalize. Hence, the referee suggestion to slightly change the focus.

Address the issue of basing estimations that assume selectively neutral markers on RNAseq data? Is there a way to assess if using coding regions (i.e., influence of selection) have an effect upon the outcome?

Best regards,

Per [Palsboll]

Reviewer 1 ·

Basic reporting

Overall, the manuscript is clearly written in professional, unambiguous language. The manuscript provide sufficient introduction and background with relevant prior literature.
The manuscript provides an important evaluation of the demographic inference of mussels when employing different sequencing techniques, sampling method, number of loci and methods of binning jSFS. However, this is not clearly reflected in the title and other parts of the text.
For example, the title is a bit misleading. You do not look at the impact of model complexity on ABC-inferred demographic history. You are inferring the demographic history of mussels among a set of competing models. To evaluate the impact of model complexity I would expect different sets of competing models in which you increase or decrease the level of complexity. In addition, to evaluate different data sets in which the “true” model would change complexity (not only mussels, in this case)
What do you mean with high-throughput sequencing strategy? To the sequencing techniques? The type of jSFS? It is very ambiguous. In the abstract, you mention sampling strategy. Are you referring to that? Then it is not sequencing strategy.
You could write a title on this line: Inferring demographic history of mussels employing different sequencing techniques (or specify the techniques here) and sampling methods (or configurations of data) as you write on line 106.

Line 106-111 You are somehow mixing the terms throughout the manuscript sampling strategy, configuration of the data, sequencing strategy. Be consistent on what do you mean with each term. You can do it here at the Introduction for example, and keep this throughout the text

Experimental design

The research is original, the knowledge gaps are identified, and the methods are well described. However, I am mainly concerned of one fundamental issue: the true or correct model is unknown.
Although the authors mentioned that the history of the species is well known, the exact model is unknown. In this sense, it is not possible to conclude that the model selected is the correct model. It is only possible to describe consistency of the results among different sample size, sequencing techniques, etc.
In addition, if the model is unknown. How can you reject the possibility of a systematic error on the estimate, if you are not contrasting the results with a “correct” and an “incorrect” model? This can be assessed by simulating data under different migration conditions, perform the ABC analysis with this data as you did with the mussels and see if you are able to recover the correct model among those models. Otherwise, you are simply inferring the demographic history of the species and not assessing the ABC method itself.

Validity of the findings

The authors show the results of the model with the highest posterior probability (Table 3) and the Bayes factor. However, I could not find the results of the other models. Results of the other models should be presented at least on the supplementary material. What does the values of Bayes factor means? Most models show Bayes factors close to 1-2. These values are usually consider low. How different are the values from the other models?

The goodness-off-fit is only shown from the most likely model. How different was compare to the goodness-off-fit from the results of the other models? You should present some examples in the supplementary material as well, or describe it in the results.

Conclusions do not emphasize clearly the main findings of the research. Conclusions should be focused on the consistency of the results and not on obtaining the correct model or making inference about the complexity of the models. For example:

Line 35. "We show that these models consistently outperform the simpler alternatives." This statement should be reformulated. Obtaining a more complex model as the most likely model does not necessarily mean that complex models outperform simpler models. They simply were selected as the most probably model in this case. You can write something like: We show that these model were consistently selected as the most probable model". You still don't know the real model of the mussels, and you don't know that if there is a species with a "true" simple migration model, the simple migration model will be selected more accurately than if there would be a species with a "true" complex models, as you assume in this case.

Line 36. "This argues that methods that are restricted to simpler models may fail to reconstruct the true speciation history." This is actually true for all models, including the "complex" models that you are evaluating.

The conclusion paragraph. Line 491-506. Reads more like an introduction. There are not clear concluding remark of the study per se.

Additional comments

The main focus of the manuscript should be modified or simulated data should be added.
You can focus on the demographic history of mussels and the consistency of the results when employing sequencing methods, sampling, etc. However, in order to make inferences about the impact of these conditions (i.e., sequencing methods, sampling, classes of jSFS) in the accuracy of the ABC inferences, you need to base your results on simulated data in which you can have control of the true model.
You can also combine simulated data with the mussels as an example, but you can not talk about the accuracy of detecting a correct model with only the mussels data.

·

Basic reporting

The manuscript by Fraisse et al. is written well, with the appropriate literature cited. The figures are of high quality and overall it's a nice contribution. My only suggestion has to to do with the conclusion; the real value I see of this study was the ABC models with novel summary statistics, small sample size, and common genomic data sets. However, the conclusion is about speciation and the current genomic islands debate; I appreciate the authors interest in the subject, but I'm not sure the conclusion is the most reflective of this ms' contribution.

Experimental design

The paper and models are appropriately designed. I was slightly interested in the use of RNAseq (i.e. coding) as I have not really seen these data used in an ABC-coalescent framework. If your model assumes random mating (as the W-F model does), then selection at these loci means the assumption of random mating is violated - would this not be the case to some extent for RNAseq data? Stronger selection yields shorter the coalescent times (assuming directional and not balancing selection) and estimates of historical Ne will be downwardly biased. As I see the main value of this paper in the idea of using RNAseq on a few samples to learn something about the population history, I just wanted to make sure it was indeed an appropriate use.

Validity of the findings

Data and models are robust - nice contribution to the genomics and demography. Again, as more of these genome ABC studies are produced, the more applications and limitations we are discovering. This is a very important contribution in that regard.

---

## Round 0.2 · Minor Revisions

Please undertake the remaining minor revisions suggested by the referee.

Best regards,

Per [Palsboll]

Reviewer 1 ·

Basic reporting

The manuscript has improved significantly; the suggestions were taken into account.

Experimental design

No comments

Validity of the findings

The aim of the study and conclusions are now well explain and consistent with the results.

Additional comments

I only have minor suggestions:
Be consistent with the use of abbreviations and only introduce the terms once. For example, the term: joint site frequency spectrum (jSFS) is introduced together with the abbreviation in line 26 of the abstract, in line 30 or the same abstract, the entire term is used again.
Similarly, you introduced joint site frequency spectrum (jSFS) in line 56 of the introduction but on the rest of the text, you mix the use of the term and the abbreviation, e.g., see lines 443, 454.

The word “our” is used on several occasions. Avoid the use of that and instead, specify what it is such as the method, the specific results, etc. For example, line 283. “We checked our ability to…” It seems that it is the ability of the authors and not the ability of the ABC method. The same in line 363. Line 416. “our capacity…”, the authors capacity or is it the capacity of the method?

Line 459. The word “confirming” is probably not the most appropriate since Robinson et al., 2014 evaluated few and large number of individuals (2-50 individuals) but this study only few (2-8). Maybe you can use other word like “consistent with”, or you can write it in a different way in which you use the results from Robinson et al., 2014 to support the results from your data.

The inferences based on the simulated data during the model checking and the inferences based on the empirical (mussel data) are now clear. I acknowledge the use of the word “simulated” to clarify. However, it is not necessary to completely remove the term pseudo-observed datasets (since it is a common term when referring to the data sets simulated for the model checking).

Line 489. Which simulation results? Clarify. Is it simulated data generated as part of the model checking ABC framework or is it additional simulated data that was independently analyzed in ABC, such as the paper you are citing in line 495 which employs simulated data generated externally using Hudson’ ms?

---

## Round 0.3 · accepted · Accept

Thank you for undertaking the revisions.

Best regards,

Per [Palsboll]

#

---

## Author Rebuttal · Round 0.3

*Institute of Science and Technology*

Institute of Science and Technology
Am Campus 1
3400 Klosterneuburg, Austria

Klosterneuburg, June 12th, 2018

Dear Prof. Per Palsboll,

We are thankful for reconsidering our work. We have made the minor changes requested by the first referee. The new version has been uploaded onto the PeerJ website, together with a 'revised tracked changes' manuscript.

With best regards,

Christelle Fraïsse
Postdoctoral FWF Fellow,
+33 (0)6 69 55 20 85
christelle.fraisse@ist.ac.at

On behalf of all authors.

# Editor's Comments

**MINOR REVISIONS**

Please undertake the remaining minor revisions suggested by the referee.

Best regards,
Per [Palsboll]

# Reviewer 1 (Anonymous)

## Basic reporting

The manuscript has improved significantly; the suggestions were taken into account.

## Experimental design

No comments.

## Validity of the findings

The aim of the study and conclusions are now well explain and consistent with the results.

## Comments for the Author

Thanks for your positive appraisal.

I only have minor suggestions:

Be consistent with the use of abbreviations and only introduce the terms once. For example, the term: joint site frequency spectrum (jSFS) is introduced together with the abbreviation in line 26 of the abstract, in line 30 of the same abstract, the entire term is used again.
Similarly, you introduced joint site frequency spectrum (jSFS) in line 56 of the introduction but on the rest of the text, you mix the use of the term and the abbreviation, e.g., see lines 443, 454.

As you recommended, we now introduce the terms *joint site frequency* spectrum and *Approximate Bayesian Computation* only once in the Abstract and in the Introduction. And we use the abbreviations *ABC, jSFS, SFS* in the remaining of the text. Models and their abbreviation were introduced in the M&M. However, we preferred to keep employing both the full model names and their abbreviation in the Results, Discussion & Conclusion as this can be useful for readers not familiars with these models. The abbreviations were mostly helpful for the Tables and Figures.

The word "our" is used on several occasions. Avoid the use of that and instead, specify what it is such as the method, the specific results, etc. For example, line 283. "We checked our ability to…" It seems that it is the ability of the authors and not the ability of the ABC method. The same in line 363. Line 416. "our capacity…", the authors capacity or is it the capacity of

the method?

*You are right, our formulations were incorrect. We reformulated accordingly along lines:*

- **268:** *We checked the ability of our ABC framework to correctly recover the true model by a "leave-one-out cross validation" from our simulations.*

- **346:** *(ii) on the ability of the method to discriminate between different speciation scenarios based on simulated datasets (Table S3).*

- **399:** *We further evaluate, by simulation, the effect of binning the jSFS on the capacity of the method to infer the correct speciation model (Table S3b).*

Line 459. The word "confirming" is probably not the most appropriate since Robinson et al., 2014 evaluated few and large number of individuals (2-50 individuals) but this study only few (2-8). Maybe you can use other word like "consistent with", or you can write it in a different way in which you use the results from Robinson et al., 2014 to support the results from your data.

*Thanks for the suggestion. We replaced* confirming *by* consistent with *on* **line 440**.

The inferences based on the simulated data during the model checking and the inferences based on the empirical (mussel data) are now clear. I acknowledge the use of the word "simulated" to clarify. However, it is not necessary to completely remove the term pseudo-observed datasets (since it is a common term when referring to the data sets simulated for the model checking).

*We agree with you, and so we put back the term* pseudo-observed datasets *in the legends of* **Figure S2** *and* **Table S3,** *and in certain places along the manuscript:*

- **line 270**: *For each of the 100 of datasets simulated under a given model (i.e., pseudo-observed datasets),*

- **line 272**: *The accuracy rate for model M was calculated as the proportion, among pseudo-observed data inferred to correspond to model M, of those actually generated under model M.*

- **line 274**: *The ambiguity rate was computed as the proportion of pseudo-observed data generated under model M whose best model was not strongly supported*

Line 489. Which simulation results? Clarify. Is it simulated data generated as part of the model checking ABC framework or is it additional simulated data that was independently analyzed in ABC, such as the paper you are citing in line 495 which employs simulated data generated externally using Hudson' ms?

*We are sorry for the confusion, we were referring to the simulated data generated as part of the model checking. We now write on* **line 470**: *Model checkings through simulations pointed out the loss of information when only four or seven classes in the jSFS were considered.*